# Pseudo Labels for Single Positive Multi-Label Learning

**Julio Arroyo**
California Institute of Technology

## Abstract

The cost of data annotation is a substantial impediment for multi-label image classification: in every image, every category must be labeled as present or absent. Single positive multi-label (SPML) learning is a cost-effective solution, where models are trained on a single positive label per image. Thus, SPML is a more challenging domain, since it requires dealing with missing labels. In this work, we propose a method to turn single positive data into fully-labeled data: *"Pseudo Multi-Labels"*. Basically, a "teacher" network is trained on single positive labels. Then, we treat the "teacher" model's predictions on the training data as ground-truth labels to train a "student" network on fully-labeled images. With this simple approach, we show that the performance achieved by the "student" model approaches that of a model trained on the actual fully-labeled images.

## 1 Introduction

In the standard *multi-class* image classification setting, the goal is to predict one applicable label for a given input. Images often contain objects of different kinds, though. So, a more realistic — and challenging — setting is *multi-label* image classification, where the objective is to predict all applicable labels for images that may belong to multiple categories. In this setting, data annotation is much more laborious because it requires labeling every category as present/absent. This has motivated work in learning multi-label classifiers with less supervision (Verelst et al., 2023), (Zhou et al., 2022), (Durand et al., 2019), (Cour et al., 2011). At the limit of minimal supervision is single-positive multi-label (SPML) learning (Cole et al., 2021): for a given image, only one category is confirmed to be present and all others are unknown. Thus, the main question in SPML is how to deal with missing labels effectively. The naive approach is to assume that unknown categories are absent ("Assume Negative" loss AN) (Cole et al., 2021), but the introduction of false negatives adds noise to the labels. A better, but still simple, approach is to maximize the entropy of predicted probabilities for all unobserved labels ("Entropy Maximization" EM loss) (Zhou et al., 2022).

Learning from incomplete labels is not a unique problem to SPML, though. In *multi-class* image classification, "Pseudo Labels" (Lee et al., 2013) — predictions treated as ground truth — have been used in situations where there is a small dataset of annotated images and a large collection of unlabeled data. In that setting, one model is trained on the small annotated dataset and is then used to synthetically annotate the large collection of unlabeled data (Yalniz et al., 2019). The large artificially-labeled dataset can then be used to re-train the same model (self-learning) or train a new model with more data resources. Like SPML, this is a setting with limited supervision. The difference however is that in SPML all of the datapoints are partially labeled (with a single-positive label), whereas in the typical "Pseudo Label" setting there is a small amount of fully annotated data and a vast amount of unlabeled data. The common goal in both, though, is to use the available data resources to recover as much supervision from the unlabeled data. In this work, we adapt the ideas of "Pseudo Labels" to SPML.

## 2 Pseudo Multi-Labels

Consider a multi-label classification problem with categories $\{1, \ldots, L\}$. Let $\{(\boldsymbol{x}_n, \boldsymbol{z}_n)\}_{n=1}^N$ be a single-positive dataset, where for input image $\boldsymbol{x}_n$ the label $\boldsymbol{z}_n \in \{1, \varnothing\}^L$ has exactly one category present $z_{ni} = 1$ and all other categories $j \neq i$ are unobserved $z_{nj} = \varnothing$.

---

**Algorithm 1** Pseudo Multi-Labels for SPML

---

**Inputs:** Single-positive dataset $\{(\boldsymbol{x}_n, \boldsymbol{z}_n)\}_{n=1}^N$, Threshold parameter $\tau \in [0, 1)$, Untrained neural networks "TeacherNet" $f(\cdot; \theta)$ and "StudentNet" $g(\cdot; \phi)$

**Output:** Trained multi-label classifier $g$.

1: Train $f(\cdot; \theta)$ on $\{(\boldsymbol{x}_n, \boldsymbol{z}_n)\}_{n=1}^N$ with some SPML algorithm.
2: **for** $n \leftarrow 1$ to $N$ **do**
3:     $\hat{\boldsymbol{y}}_n \leftarrow f(\boldsymbol{x}_n; \theta)$
4:     **for** $i \leftarrow 1$ to $L$ **do**
5:         $\tilde{y}_{ni} \leftarrow \begin{cases} 0 & \hat{y}_{ni} \leq \tau \\ 1 & \hat{y}_{ni} > \tau \end{cases}$  ▷ Make "Pseudo Multi-Labels" from "TeacherNet" predictions
6:     **end for**
7: **end for**
8: Train $g(\cdot; \phi)$ under full-supervision with the "Pseudo Multi-Labels" $\{(\boldsymbol{x}_n, \tilde{\boldsymbol{y}}_n)\}_{n=1}^N$

---

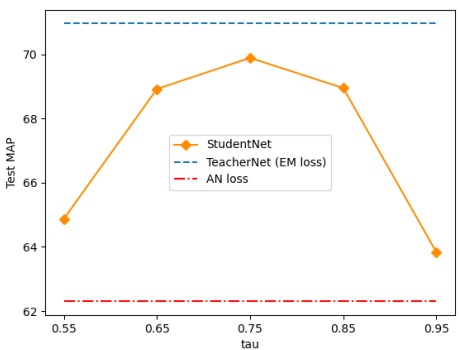

(a) Test set mean average precision (MAP)

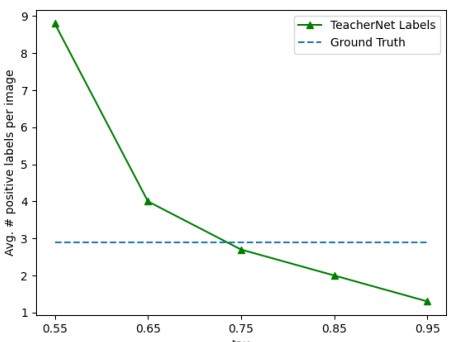

(b) The threshold $\tau$ defines a level of strictness on "TeacherNet's" predictions, and thus it controls the average number of positive labels per image on the "Pseudo Multi-Labels" dataset.

## 3   EXPERIMENTS

Following the experimental setup of Cole et al. (2021), we produced a single positive version of the COCO dataset (Lin et al., 2014), by picking one positive label per image uniformly at random and discarding the rest. We kept the validation and test sets uncorrupted. We ran Algorithm 1 for five evenly spaced values of $\tau \in [0.55, 0.95]$: a higher value of $\tau$ indicates that "TeacherNet" must make a high confidence prediction for a category in order for it to be turned into a positive label. So, $\tau$ is an important hyperparameter for our method. In Figure 1b, we show how $\tau$ controls the average number of positive labels per input image. In Figure 1a, we present the final model's performance compared to the baseline "Assume Negative" loss and state-of-the-art "Entropy Maximization".

## 4   CONCLUSION

One of the main challenges with our method is that the hard cutoff $\tau$ on "TeacherNet's" predictions when generating the "Pseudo Multi-Labels" may produce noisy labels. This is also the problem with the "Assume Negative" model, but in our proposed method it happens to a lesser extent. So, although our method beats the baseline "Assume Negative" model, it does not beat the state-of-the-art.

### URM STATEMENT

The authors acknowledge that at least one key author of this work meets the URM criteria of ICLR 2023 Tiny Papers Track.

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
