# OpenReview forum: "Pseudo Labels for Single Positive Multi-Label Learning"
_ICLR.cc/2023/TinyPapers — Submitted to Tiny Papers @ ICLR 2023_

### Official Review · Reviewer_CNhq · 2023-03-27

**Confidence:** 4

**Summary Of Contributions:**

The paper proposes a method called Pseudo Multi-Labels for multi-label image classification using single positive data. The authors use a "teacher" network trained on single positive labels to predict ground truth labels for a "student" network trained on fully-labeled images.

**Rating:**

Clear, Correct, and Reproducible (CCR): a submission which meets the reviewing criteria

**Strengths And Weaknesses:**

Strengths:
- The paper addresses an important problem in multi-label image classification: the cost of data annotation.
- Pseudo Multi-Labels is a simple yet effective approach to turning single positive data into fully-labeled data.
- The paper provides a clear explanation of the proposed method and its implementation.
- The experimental results demonstrate that Pseudo Multi-Labels outperforms the baseline "Assume Negative" model.

Weaknesses:
- In the abstract, it is mentioned that the proposed method achieved a model trained on fully-labeled images, but there was no results. It would be helpful to include the results in Figure (a).
- It seems necessary to compare the performance of the best model tuned with $\tau$ using the validation set. If we can find the best $\tau$ with the validation set, the proposed method would be even better.


**Suggested Changes:**

Suggested changes:
- I wonder if the performance of the student network is lower than that of the teacher network even when using a teacher network trained with another losses (ex. AN loss), since the performance of the student network was lower than that of the teacher network when trained with the EM loss. If the same phenomenon is found, we may be able to determine that the teacher-student structure is not effective in the SPML problem.
- I recommend reading the following paper as related work. It may help in extending this study: "Large Loss Matters in Weakly Supervised Multi-Label Classification (CVPR 2022)."

Minor:
- The figure does not have a caption. Please add a caption.

---

### Official Review · Reviewer_FJa6 · 2023-03-29

**Confidence:** 4

**Summary Of Contributions:**

The paper studies the single positive multi-label (SPML) learning where the models are trained on a single positive label per image. The authors propose a method called "Pseudo Multi-Labels" to convert single positive labeled data into fully-labeled data for multi-label image classification. The method involves training a "teacher" network on single positive labels and using its predictions as ground-truth labels to train a "student" network on fully-labeled images.

**Rating:**

Clear, Correct, and Reproducible (CCR): a submission which meets the reviewing criteria

**Strengths And Weaknesses:**

Strong aspects:

1. The paper clearly identifies the setting of single positive multi-label (SPML) learning and its differences with multi-class and vanilla multi-label classification.

2. The paper presents an approach called "Pseudo Multi-Labels" to turn single positive data into fully-labeled data for training multi-label classifiers. A "teacher" network is used to predict labels for single positive data, which are then used as groundtruth labels to train a "student" network on fully-labeled images.

3. The paper provides experimental results demonstrating that the proposed approach is better than “Assume Negative” model, but it does not beat the state-of-the-art.

**Suggested Changes:**

The paper does not provide a detailed explanation of the technical aspects of the proposed approach, such as the architecture of the "teacher" and "student" networks. Such information might be beneficial to the community.

---

### Author Response · Authors · 2023-05-29
**Archival**

We wish to opt-in for archival.

---

### Meta-Review · Area_Chair_wxT4 · 2023-04-06

**Recommendation:** Invite to present
**Confidence:** 5

**Metareview:**

This paper tackles the single positive multi-label image classification setting, and proposes a method that leverages a teacher-student framework where the teacher is trained on single positive labels and the prediction of the teacher is used as the pseudo-label for the student.

The reviewers all agree that this paper is clear, correct, and reproducible, and that this paper presents a clear explanation of the proposed method and its implementation.
Some minor concerns are raised, and the authors are encouraged to consider these concerns in a revision, such as: the caption of the figures and including more results.


**Summary:**

A clear presentation of a teacher-student framework for the task of single positive multi-label image classification.

**Reason For Not Giving A Higher Recommendation:**

This paper is clear, but as reviewer CNhq noted, some revisions are needed.

**Reason For Not Giving A Lower Recommendation:**


All reviewers found this paper to be CCR and meet the review criteria

---

### Decision · Program_Chairs · 2023-04-10

Invite to present